# Categorical consistency of parity and magnitude facilitates implicit learning of color-number associations

**Talia L. Retter** [1,2]*, **Christine Schiltz**[1]

**1** Department of Behavioural and Cognitive Sciences, Institute of Cognitive Science & Assessment, University of Luxembourg, Esch-sur-Alzette, Luxembourg, **2** Université de Lorraine, CNRS, IMoPA, Nancy, France

* talia.retter@univ-lorraine.fr

## Abstract

Perceiving high-frequency stimulus pairings may lead to implicit associative learning. Interestingly, category-level pairings, such as blue-even, may facilitate implicit learning relative to item-level pairings, such as blue-2 and blue-7. Such an advantage of categorical consistency has been previously demonstrated for associative learning with parity; here, we replicate this finding, and extend it to a second, more-often studied category, magnitude. In a parity experiment, participants reported the parity of single-digit numerals; numerals appeared in either blue or yellow, but throughout, participants were not given any information about color. In the novel magnitude experiment, the same participants reported the magnitude of single-digit numerals appearing in either purple or green. Associative learning was assessed through the comparison of response performance to congruent (high-frequency color-number parings; p = .9) vs. incongruent (low-frequency; p = .1) trials. A robust congruency effect was found at the category-level for both parity (accuracy: 8%; response time (RT): 54 ms) and magnitude (accuracy: 4%; RT: 37 ms), but not at the item-level. A third, novel parity-mix experiment, with purplish-blue and greenish-yellow, was also tested with these participants, in order to probe for potential interactions of colors associated across parity and magnitude dimensions. There was a congruency-effect advantage for parity-magnitude matching numerals vs. mismatching in terms of accuracy (4%), suggesting that color associations with conceptual categories may relate to each other. An explicit association report task revealed above-chance accuracy for the color of numerals for both parity and magnitude at the category-level, and for parity at the item-level. These results suggest that categorical consistency of multiple numerical concepts may facilitate implicit learning of both specific and multidimensional color-number associations.

**Data availability statement:** Data are publicly available on the Zenodo database: https://doi.org/10.5281/zenodo.16024168.

**Funding:** This work was supported by structural funding allocated by the Faculty of Humanities, Education and Social Sciences of the University of Luxembourg (FHSE/UL; https://www.uni.lu/fhse-en/; to CS at the CNSlab/EPSYLON) and a postdoc fellowship from the France 2030 program Initiative d'Excellence Lorraine (LUE; https://www.univ-lorraine.fr/lue/; to TLR as part of ANR-15-IDEX-04-LUE).

**Competing interests:** The authors have declared that no competing interests exist.

## Introduction

Associative learning, in which covarying stimuli or events form linked representations, naturally shapes our perceptual experience of the environment. Indeed, associative learning may reflect contextual framing of stimuli in the environment, serving to simplify and facilitate our understanding of an otherwise overwhelming amount of available sensory information (e.g., [1–3]). In this perspective, associative learning can be seen as a lens through which information about the world is filtered automatically to benefit perceptual processing. Indeed, associative learning does not appear to rely on our awareness of rules in the environment: associative learning may occur both *implicitly,* i.e., without any conscious awareness and in the absence of explicit instruction [1,4–8], and *statistically,* across covarying stimuli or events that occur together not every single time, but probabilistically, with an above-chance likelihood [9–12].

Implicit, statistical associative learning may be probed through comparing behavioral responses to one stimulus or feature, such as color, that is presented with a high-probability in conjunction with another stimulus or feature, such as a shape, symbol, or word (e.g., [11,13–18]). Such associative learning has been best evidenced at the item-level [19–21]. At this level, individual (or small chunks of) stimulus features are thought to be associated with other features at the sensory and/or response level, without general abstractions. Associative learning at the category-level has been examined less, in only a few studies with color associations to our knowledge ([17,18,22–24]; but in the context of sequence order regularities: [25–27]; priming: [28,29]; and visual search: [30]). In the sense that associative learning is assumed to facilitate understanding of extensive incoming perceptual information, it is logical that such a conceptual filtering would be more efficient than an individual case-by-case one, such that this topic warrants further investigation.

In a previous study, we used the category of numbers' parity to demonstrate an advantage of categorical consistency for implicit learning of color-number associations [18]. In that study, single-digit numerals were presented in different font colors: color was consistent with parity in a category-level version of the experiment (2,4,6,8 high-probability blue; 3,5,7,9 high-probability yellow); in a control item-level version of the experiment with different participants, the numbers were not categorically grouped with the high-probability color associations (blue: 2,3,6,7; yellow: 4,5,8,9). That is, in the "category-level" version of the experiment, numerical (even/odd) and color (blue/yellow) categories were consistently paired. In using color and number stimuli, it was possible to consistently control the quality of sensory information (8 numbers paired with 2 colors) across the category- and item-levels. Participants' task was to report the parity of each number, and participants were not given any information about color. Associative learning was assessed by comparing the response performance advantage for congruent (high-frequency color-number parings) to that of incongruent (low-frequency) trials. A robust effect of associative learning was evidenced at the category-level (8.3% accuracy and 40 ms response time; RT), but not at the item-level (effects of −1.5% accuracy and 9 ms RT).

Here, our first aim was to replicate these results for parity: that categorical consistency facilitates implicit learning. Parity is an interesting category for studying category-level learning, since parity is essentially binary, enabling a simple division of numbers into even and odd categories that each may be associated with a different color category (although some numbers may appear more strongly even or odd than others, e.g., [31]). It has been questioned whether parity is processed automatically: this could be explored with this paradigm by reversing the task (to an explicit color task) and investigating whether color-parity associations were still learned implicitly ([24]: see the following paragraph). However, magnitude is the numerical property of primary interest in numerical cognition research (e.g., [32–34]), as well as in color coding of educational materials (a potential application of this research; e.g., [35]).

Therefore, our second aim was to extend the category-*vs.*-item-level learning paradigm to test whether implicit category-level magnitude associations may occur with color, despite magnitude being a more continuous dimension. Magnitude associations with color, in addition to parity associations with color, were tested previously in a reversal of this paradigm, in which the task was to report each number's color [24]. In that version of the paradigm, participants were not given any information about numbers, and associative learning was still evidenced significantly at the category-level for both magnitude and parity (magnitude: 1.3% accuracy congruency effect; 9 ms RT; and parity: 3.1% accuracy), but not at the item-level. As mentioned above, the color task in that design, which required participants to report whether numbers were blue/yellow in the parity experiment and red/green in the magnitude experiment, enabled probing whether the numerical properties of parity and magnitude were automatically elicited from symbolic numerals (see [34,36–38]). In that theoretical context, the primary category under investigation was parity, for which the automatic processing has been more contentious: magnitude was used mainly as a reference dimension.

The third and final aim of the present study was to test the hypothesis that there might be an interaction between colors associated with parity and magnitude, predicting the best associative learning for parity-magnitude consistent colors. While color associations in the context of long-term memory are thought to be specific (e.g., in the case of color-associated objects, such as yellow with a banana: [39,40]), there is evidence that novel color associations learned in an experimental context may extend to perceptually different color exemplars, even possibly cross-categories [17]. Statistically learned color associations have also been proposed to occur multidimensionally, at both object- and feature-based levels [41]. In the present experiment, following parity and magnitude experiments with blue/yellow and purple/green number colors, respectively, the same participants repeated the parity experiment with purple-shifted blue (purplish-blue) and green-shifted yellow (greenish-yellow). The largest congruency effects were predicted at the category-level for colors that matched both parity and magnitude associations (e.g., purplish-blue for large-even numbers, matching both large-purple and blue-even associations). Smaller congruency effects were predicted for mismatching parity-magnitude associations (e.g., purplish-blue for small-even, mismatching large-purple associations). Altogether, these experiments test whether categorical consistency of multiple concepts may facilitate implicit learning of both specific and multidimensional associations.

## Methods

The methods closely followed those of previous studies on color-number associations [18,24], inspired by earlier color-association implicit learning paradigms (e.g., [11,14,15,17,22,23]; and the Stroop interference paradigm ([42,43]; including at the semantic level, for color-associated words: [44,45]).

## Participants

Participants were recruited from a university community for a 1.5-hour computerized and written study in cognitive neuroscience, with monetary compensation (15 euros), and if applicable, student participation credits. Forty participants took part in the experiment (20 male), with ages ranging from 18–39 years old (M = 26 years old). No participants were excluded from the data analysis. Sample size was determined relative to previous studies on implicit associative learning,

most relevantly that of [18]: robust effects with 32 participants (16 per group). All reported normal or corrected-to-normal visual acuity, no abnormalities in color perception, and no synesthesia or other learning disabilities (inclusion criteria shared during study recruitment). Two were left-handed. The most frequent main language of math acquisition was German (17 participants), followed by English (11), and French (4). All had completed at least a secondary-school level of education, with some holding bachelor's (12) and master's (8) degrees. Participants' Tempo-Test Rekenen (TTR) scores, rapidly assessing mathematical ability [46], ranged from 90–193 (out of 200; M = 140; SD = 24.9). Participants were randomly divided into experimental and control (item-level) groups, completing the category-level or item-level experiment versions, respectively: there was no difference between TTR scores for the experimental (M = 144; SD = 20.6) and control (M = 136; SD = 28.7) groups, $t_{38}$ = 0.96, p = .35, d = 0.30. Each participant was tested individually at a university laboratory, following signed, informed consent, under procedures approved by the internal university Ethical Review Panel (ERP), consistent with the Code of Ethics of the World Medical Association (Declaration of Helsinki, 2013).

### Stimuli and procedure

The experiment was prepared and presented with PsychoPy3 v2020.2.8 [47], operating on Python (Python Software Foundation). The stimuli consisted of the Arabic numerals 2–9. They were presented in Arial font, in one of the following possible colors (R/G/B 0:255): blue (0/0/255); purplish-blue (80/0/255); yellow (255/255/0); greenish-yellow (230/255/0); green (0/255/0); purple (100/0/200); and black (0/0/0; see Fig 1a). The greenish-yellow and purplish-blue colors were created to mix parity- and magnitude-associated colors, by shifting yellow towards green, and blue towards purple (while

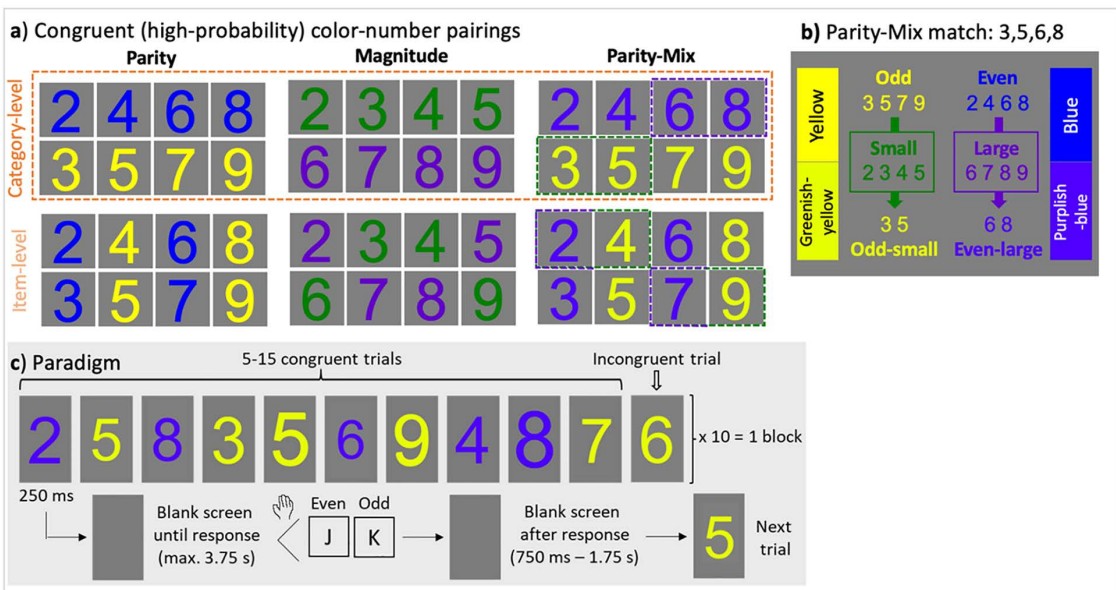

**Fig 1. Experimental design. a)** The congruent, i.e., high-probability, color-number pairings for each part of the experiment: Parity, Magnitude, and Parity-Mix. Only at the category-level, these parings are consistent with a numerical category (parity or magnitude). Implicit associative learning was assessed through the congruency effect, in which the performance for congruent trials was predicted to be better than for incongruent trials (when the number appeared in the other color of that experiment part). The match color-number pairings are outlined in the Parity-Mix experiment. **b)** Parity-Mix colors were shifted away from yellow and blue, towards green and purple, respectively. A category-level match across parity and magnitude color associations was thus generated only for the numbers 3,5,6, and 8; the numbers 2,4,7, and 9 were mismatching. **c)** An example of the trial design for the implicit color-number association learning paradigm, for the Parity-Mix category-level version of the experiment. This timing of trial presentation was consistent across the Parity, Magnitude, and Parity-Mix experiment parts, as well as in the explicit association report task (wherein numbers were presented in black).

maintaining the categories of yellow and blue that potentially will have already been associated with parity; Fig 1b). The colored number stimuli were displayed in the center of a uniform gray (128/128/128) screen (Dell S2419HGF monitor; connected to a Dell PC employing a GeForce 1050 (Nvidia) graphics card). Their average height was 3.1° of visual angle, which the size randomly varied in seven steps from 80–120% percent at each stimulus presentation, to avoid low-level perceptual image repetition effects.

The computerized experiment was structured into five parts. The first three parts, the Parity (1), Magnitude (2), and Parity-Mix (3) experiments, consisted of an *implicit color-number association learning paradigm* (Fig 1c; for other illustrations of the design, please see Fig 1 of [18]; and Fig 1 of [24]). In this design, there were 5 blocks, each consisting of 10 cycles of 5–15 consecutive presentations of congruent trials, followed by a single incongruent trial (defined below). That is, there was an average of 10 incongruent trials and 100 congruent trials per block, leading to an average of 50 incongruent trials and 500 congruent trials per experiment. Each trial consisted of a 0.25 s number stimulus presentation duration, followed by a blank screen until a response key was pressed or a maximum of 3.75 s, after which the blank screen continued for another 0.75–1.75 s, followed by the next trial. Each of the three implicit association learning experiments lasted about 15 minutes.

Participants' task was to report, using the indicated response keys, the targeted numerical property of presented numerals, as accurately and as fast as possible: parity in the Parity and Parity-Mix experiments; and magnitude in the Magnitude experiment. Specifically, in the Parity and Parity-Mix experiments, participants were instructed to use the first two fingers of their right hand to press on the response keys J and K, for even and odd, respectively. In the Magnitude experiment, participants' task was to report whether numbers were small (2–5) or large (6–9), by using the first and second fingers of their lefts hands to press on the response keys D and E, for small and large, respectively. Response keys were not counter-balanced across participants for simplicity, given that no response key biases were predicted, and to exactly replicate [18]. Different response keys and hands were used across the Parity/Parity-Mix and Magnitude experiments, in order to restrict potential response location/motor learning transfer across these experiments.

Numerals were presented in different font colors: yellow and blue in the Parity experiment; green and purple in the Magnitude experiment; and greenish-yellow and purplish-blue in the Parity-Mix experiment (Fig 1a). Critically, participants were not given any information or task related to number color, but numbers were presented with high-frequency in one color, termed the "congruent" number-color pairing; the other color was termed the "incongruent" number-color pairing (defined specifically in the following paragraph). Specifically, there was a 10:1 average ratio of congruent to incongruent color presentations, forming the basis of potential statistical learning of the color-number relationships. Performance was predicted to be high for congruent trials: if there was learning of color-number associations, this was predicted to cause interference for incongruent trials and/or improvement of performance for congruent trials, leading to selectively lower accuracy and higher response time (RT) for incongruent than congruent trials. Therefore, *congruency* was the critical variable for assessing potential implicit learning effects.

There were two levels (item-level and category-level) within each experiment, divided across participant groups (Fig 1a). In the item-level version, in which a set of four non-conceptual numbers (e.g., 2,3,6, and 7 in the Parity experiment) was congruent with one color (blue), and the set of remaining non-conceptual numbers (e.g., 4,5,8, and 9) was congruent with a different color (yellow). In the item-level Magnitude experiment, the numbers 3,4,6, and 9 were congruently paired with green, and the numbers 2,5,7, and 8 were congruently paired with purple. The category-level version was of primary interest, in which numbers sets were defined conceptually. In the Parity experiment, for the category-level version, even numbers 2,4,6, and 8 were congruent with blue, and the odd numbers 3,5,7, and 9 were congruent with yellow (Fig 1a: left). In the category-level Magnitude experiment, the small numbers 2,3,4, and 5 were congruently paired with green, and the large numbers 6,7,8 and 9 were congruently paired with purple (Fig 1a: middle).

In the Parity-Mix experiment, the Parity experiment was replicated, except that the colors blue and yellow were replaced by the colors purplish-blue and greenish-yellow, respectively (Fig 1a: right), in order to probe the potential

interaction between colors associated with parity and magnitude. Specifically, we aimed to test whether numbers that were a *match* for both parity and magnitude associations led to stronger associative learning effects relative to *mismatch* numbers (Fig 1b; outlined in Fig 1a: right). Since even numbers were high-probability associated with blue, and large numbers with purple, two purplish-blue digits were a match for both these associations (6 and 8). Similarly, since odd numbers were high-probability associated with yellow, and small numbers with green, two greenish-yellow digits were a match for both these associations (3 and 5). The remaining numbers were a mismatch across parity and magnitude color associations (2,4,7, and 9). At the item-level, green-yellow and purple-blue matching numbers (2,4,7, and 9; *vs.* mis-matching 3,5,6, and 8) could also be defined.

These first three implicit color-number association learning experiments were followed by two *explicit association report task* experiments (e.g., [15,18]): Parity explicit (experiment 4) and Magnitude explicit (experiment 5). In these experiments, numbers were presented in black, and participants' task was to report whether the number was associated more with one of two colors, as accurately and as fast as possible (same trial design as in Fig 1c, as for the implicit color number association learning paradigm; each of the numbers were presented three times, in a blocked randomization, for a total of 24 trials). In the Parity explicit report task, participants were instructed to use the second and third fingers of their left hand to report whether the number was associated more with blue or yellow, using the response keys H and T, respectively. In the Magnitude explicit report task, participants were instructed to use the second and third fingers of their right hand to report whether the number was associated more with purple or green, using the response keys C and F, respectively. That is, participants were instructed to use different response keys and a different hand to give responses for parity or magnitude in the explicit report task than in the implicit associative learning task, to probe whether potential associative learning effects extended beyond response-level learning within each (parity or magnitude) experiment type (e.g., [16]).

The order of these experiments was fixed for all participants: 1) Parity; 2) Magnitude; 3) Parity-Mix; 4) Magnitude explicit; and 5) Parity explicit. This order was imposed for several reasons: first, the implicit color-number association learning paradigm (experiments 1–3) had to precede the explicit association report task (experiments 4–5), to avoid prompting explicit learning in the implicit experiments. Second, since the Parity-Mix experiment was designed to target interactions between learned parity and magnitude implicit color associations, this experiment had to appear third: in order that this experiment was not immediately adjacent to the original Parity experiment that it replicated with shifted colors, potentially impacting participants' perception of the colors, the Parity experiment was always presented first. Finally, since the implicit learning paradigms always ended with Parity-Mix, the explicit report always tested Magnitude explicit first, in order to maintain consistent recency of parity and magnitude reports within the testing session and across participants.

After completion of the computerized experiment, the participants completed a written 3-question post-experiment questionnaire (as in [24]; but not yet introduced in [18]), assessing potential implicit learning of color-number associations. The first question was designed to enable open observations, without any specific prompting: 1) What did you notice about the experiment? The second question specifically asked about color-number associations: 2) Did you notice that certain numbers usually appeared in a certain color? If so, please give a description. Finally, we wanted to assess participants' memory about the colors used in the experiment, since these colors were not implicated in participants' given experiment information or task instruction: 3) What colors did you see during the experiment? Block 1: parity task; Block 2: magnitude task; Block 3: parity task. We were also interested in whether participants would notice the shift of colors in the Parity-Mix experiment, or if any participants might have observed these as purple or green. Finally, participants performed a written, 4-minute TTR task assessing mathematical ability ([46]; see Participants).

## Data analysis and statistics

Behavioral responses in the implicit color-number association learning paradigm were processed in terms of accuracy and response time (RT), using Excel Version 2402 (Microsoft). Response time was assessed only for correct trials, and any responses ±2.5 standard deviations from each individuals' mean were excluded (M = 2.7% of trials; SD = 0.807%; range

across participants: 0.64% to 4.56%). Congruency (congruent vs. incongruent trials) was used to assess potential associative learning effects, across experiment levels (category-level vs. item-level), separately for each the Parity, Magnitude, and Parity-Mix experiments. Repeated-measures analyses-of-variance (RM ANOVAs) were conducted with SPSS Statistics 28 (IBM) using the within-participants factor of *Congruency* and the between-participants factor of *Level.* Follow-up paired-samples t-tests were applied to test for congruency effects within each experiment level (one-tailed, predicting incongruent < congruent accuracy; and incongruent > congruent RT). The explicit association report task accuracy was analyzed with one-sample t-tests, one-tailed against 50% chance-level; to compare across experiment levels, one-tailed, independent-samples t-tests were used. In the Parity-Mix experiment, to assess an effect of *Match* at the category-level (match: 3,5,6,8 vs. mismatch: 2,4,7,9), one-tailed, paired-samples t-tests were conducted, predicting match > mismatch congruency effect performance; one participant was excluded from this analysis for having an accuracy congruency effect over 2.5 SDs from the mean. Two participants did not provide any responses in the explicit association report task for Parity, and three for Magnitude, at the category-level, presumably due to lack of task comprehension. In the event that Levene's Test for Equality of Variances was violated, corrected degrees of freedom were applied.

## Results

### Parity

The results demonstrated large congruency effects for parity only in the category-level version of the experiment (Fig 2a, 2b; dataset: [48]). At the category-level, there was a 7.9% congruency effect (congruent – incongruent; congruent: M = 94.6%; SE = 0.91%; incongruent: M = 86.7%; SE = 2.2%), and a 53.9 ms RT effect (incongruent – congruent; congruent: M = 545.0 ms; SE = 15.7 ms; incongruent: M = 598.9 ms%; SE = 17.3 ms), while at the item-level the congruency effects were very small: −0.1% accuracy (congruent: M = 95.1%; SE = 0.63%; incongruent: M = 95.2%; SE = 1.0%) and 5.4 ms RT (congruent: M = 578.9 ms; SE = 22.7 ms; incongruent: M = 583.6 ms; SE = 21.4 ms).

There was a significant interaction between *Congruency* and *Level,* for both accuracy, $F_{1,38} = 12.3$, $p = .001$, $\eta_p^2 = 0.25$, and RT, $F_{1,38} = 25.5$, $p < .001$, $\eta_p^2 = 0.40$. Follow-up t-tests indicated significant effects of congruency in the category-level version of the experiment for both accuracy, $t_{19} = 3.73$, $p < .001$, $d = .83$, and RT, $t_{19} = 7.57$, $p < .001$, $d = 1.69$; but not in the item-level experiment: accuracy, $t_{19} = -0.098$, $p = .46$, $d = -0.022$; RT, $t_{19} = 0.71$, $p = .25$, $d = 0.16$.

### Magnitude

As for Parity, there was a substantial congruency effect only in the category-level version of the experiment (Fig 2c, 2d). At the category-level, the congruency effect was of 4.4% (congruent: M = 94.8%; SE = 0.75%; incongruent: M = 90.4%; SE = 1.73%) and 36.7 ms (congruent: M = 547.6 ms; SE = 15.6 ms; incongruent: M = 584.3 ms; SE = 18.8 ms), while at the item-level it was −1.2% (congruent: M = 95.4%; SE = 0.59%; incongruent: M = 96.6%; SE = 0.54%) and −9.1 ms (congruent: M = 604.0 ms; SE = 27.8 ms; incongruent: M = 594.9 ms; SE = 24.0 ms).

There was a significant interaction between *Congruency* and *Level,* again for both accuracy, $F_{1,38} = 10.63$, $p = .002$, $\eta_p^2 = 0.22$, and RT, $F_{1,38} = 16.7$, $p < .001$, $\eta_p^2 = 0.31$. The congruency effect was significant within the category-level version of the experiment in terms of both accuracy, $t_{19} = 2.66$, $p = .008$, $d = 0.60$, and RT, $t_{19} = 4.58$, $p < .001$, $d = 1.02$. At the item-level, the congruency effect was not significant for RT, $t_{19} = -1.16$, $p = .13$, $d = -0.26$, but it was for accuracy, in the opposite direction as predicted, $t_{19} = -2.45$, $p = .012$, $d = -0.55$.

### Parity and Magnitude continued

To demonstrate a lack of accuracy-speed tradeoffs, data were plotted at the individual level for the congruency effects in both variables (Fig 3). This demonstrated a robust congruency effect in both variables for participants at the category-level for both parity and magnitude, while item-level accuracy and RT both appeared to cluster around zero. The congruency effects were observed to evolve across blocks for the category-level version of the experiment, e.g., with a trend of

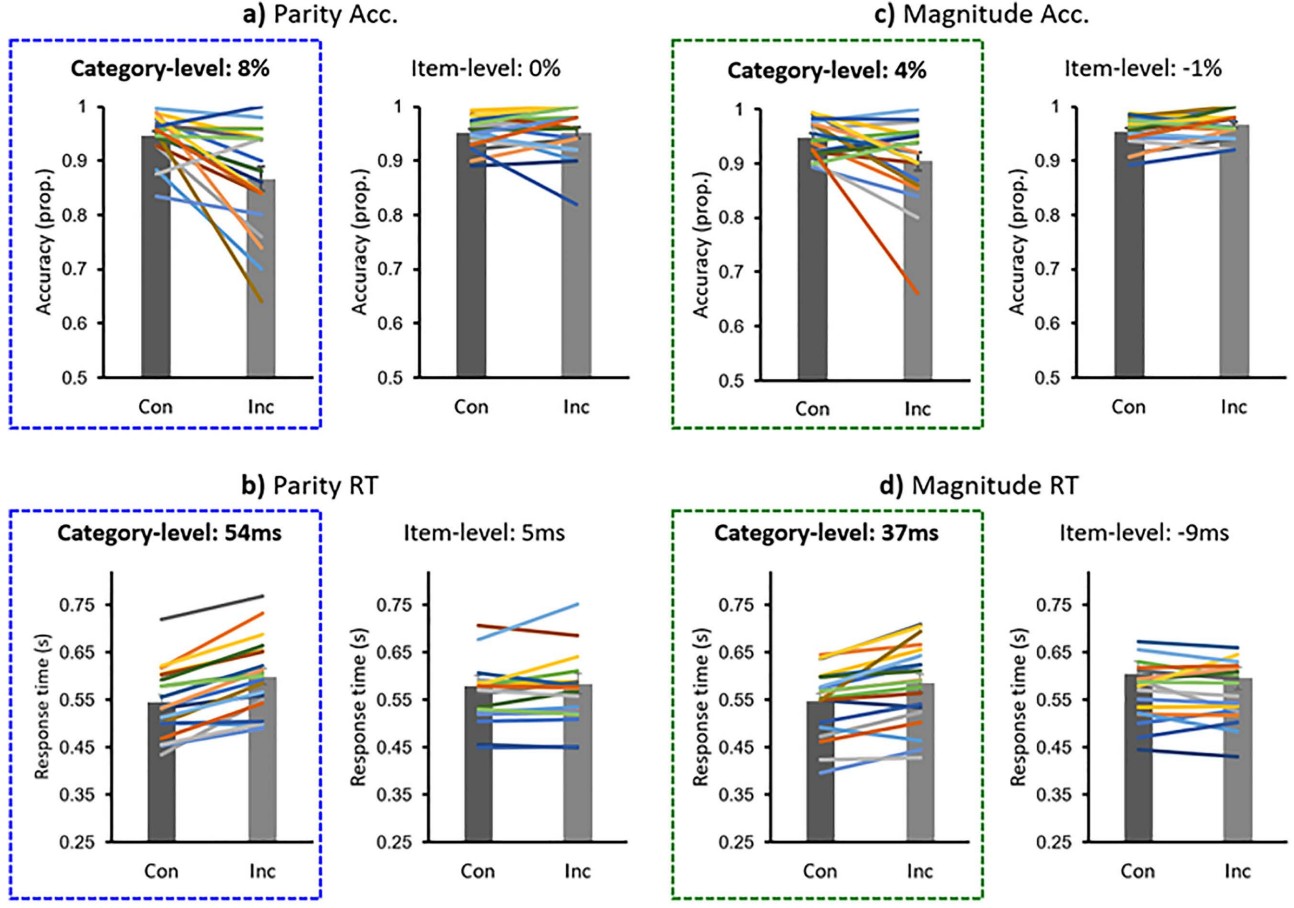

**Fig 2. Implicit color-number association learning experimental results for Parity and Magnitude: congruent (con) vs. incongruent (inc) trials.**
**a)** Parity accuracy (acc.). **b)** Parity response time (RT). **c)** Magnitude accuracy. **d)** Magnitude RT. Columns indicate mean values, with error bars of ±1 SE of the mean; colored lines indicate individual participant data.

decreasing incongruent accuracy across blocks, with little change over time for the item-level experiment, for both parity and magnitude (S1 Fig).

## Explicit Parity and Magnitude report

In terms of the explicit association report task performance (Fig 4), the category-level accuracy was above chance-level (50%) for both parity (M=66.9%; SE=4.22%) and magnitude (M=57.8%; SE=4.47%). These were both significant differences: Parity: $t_{18}$=4.00, p<.001, d=0.92; Magnitude: $t_{17}$=1.77, p=.047, d=0.42. The item-level accuracy was not significantly above chance for magnitude (50.7%), $t_{19}$=0.24, p=.41, d=0.054, but it surprisingly was for parity (63.1%), $t_{19}$=3.42, p=.001, d=0.77. Despite these above-chance accuracies, only a single participant had perfect accuracy for every item, in the category-level Parity experiment: across all experiments and levels, the majority of participants (>70%) had perfect accuracy for only 0–4 items (out of 8; Fig 4e, 4f).

## Parity-Mix

The Parity-Mix results replicated the Parity results (Fig 5a, 5b); category-level congruency effect: 6.8% (congruent: M=94.1%; SE=0.84%; incongruent: M=87.3%; SE=1.5%) and 44.9 ms RT (congruent: M=531.1 ms; SE=14.3 ms;

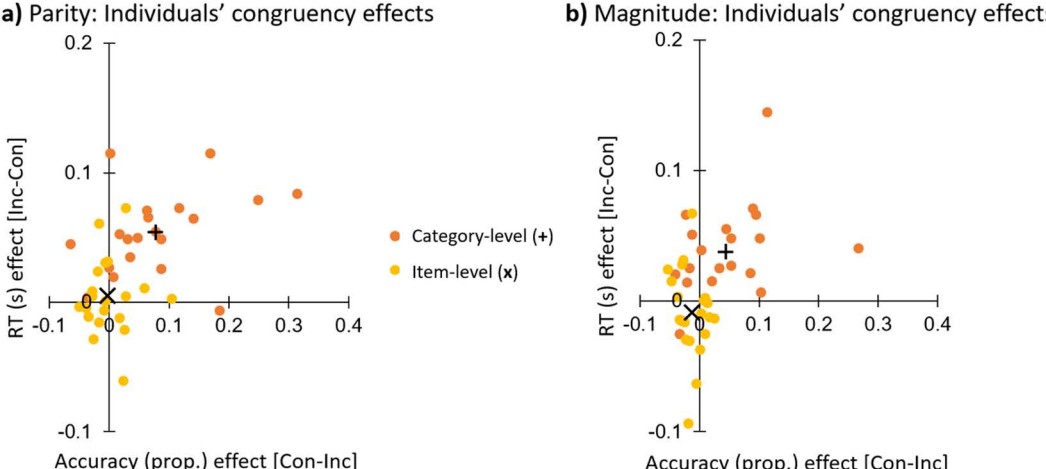

**Fig 3. Individuals' congruency effects for the Parity (a) and Magnitude (b) experiments.** Data in the upper right quadrant reflect an effect in the predicted direction for both accuracy and response time (RT). Con = congruent; Inc = incongruent.

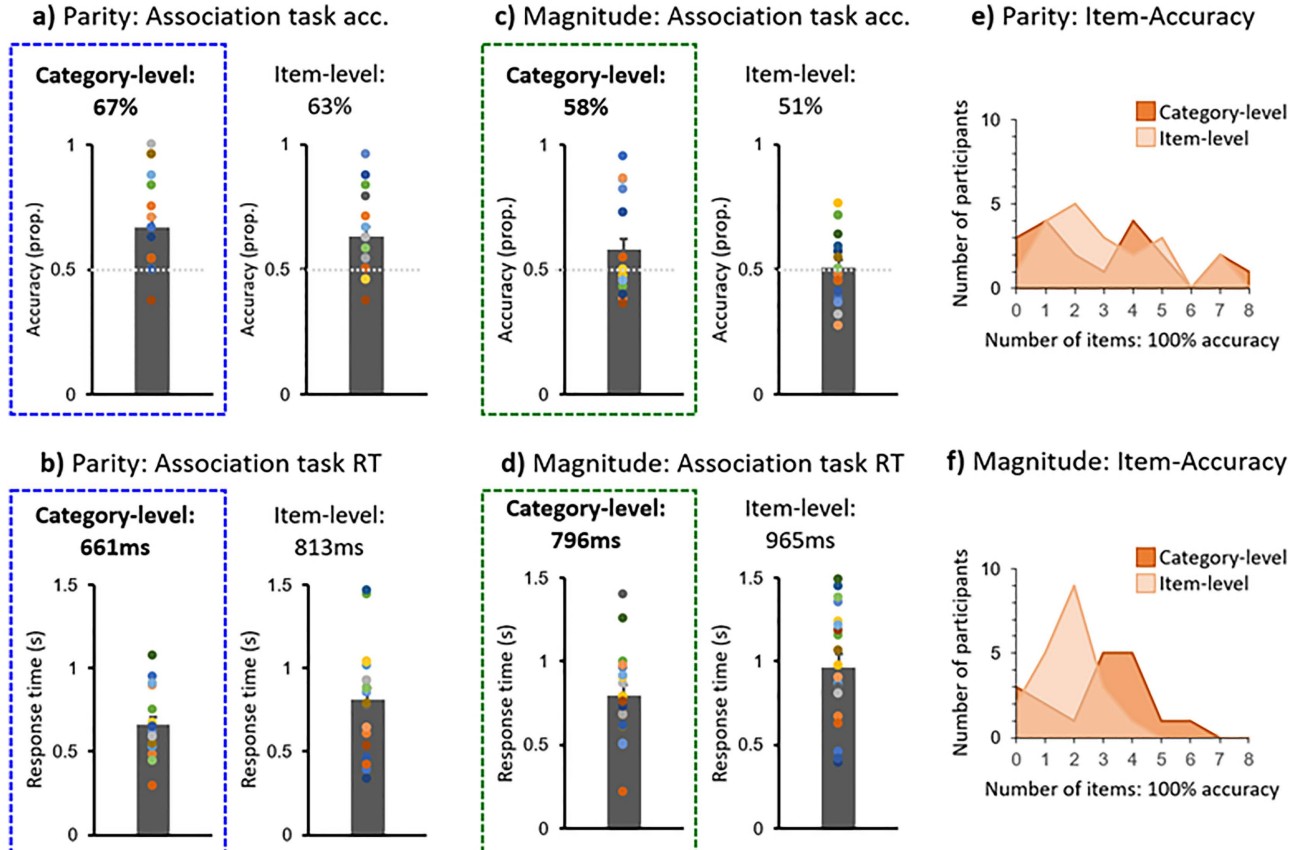

**Fig 4. Explicit association report task performance in terms of accuracy (acc.) for Parity (a) and Magnitude (b), as well as response time (RT) for Parity (c) and Magnitude (d).** Histograms of the number of participants with perfect accuracy per number of items are also plotted for Parity (e) and Magnitude (f).

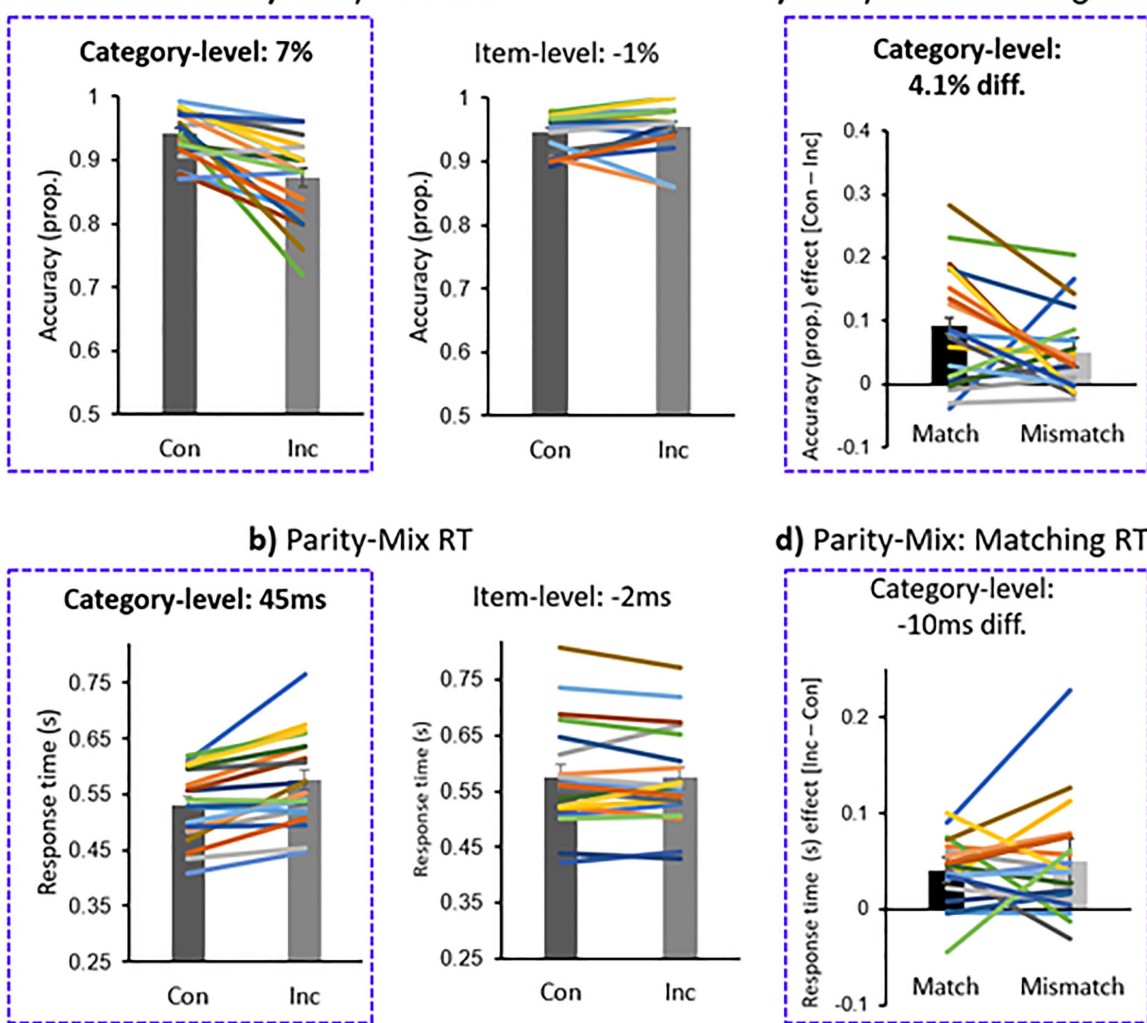

**Fig 5. Parity-Mix experimental results: congruent (Con) vs. incongruent (Inc) trials. a)** Accuracy (acc.). **b)** Response time (RT). **c)** The significant Acc. congruency effect for Match (3,5,6,8) vs. Mismatch (2,4,7,9) trials. **d)** The non-significant RT congruency effect for Match vs. Mismatch trials.

incongruent: M = 575.9 ms; SE = 18.0 ms); item-level congruency effect: −0.8% accuracy (congruent: M = 94.6%; SE = 0.63%; incongruent: M = 95.4%; SE = 0.84%) and −1.8 ms RT (congruent: M = 576.1 ms; SE = 21.4 ms; incongruent: M = 574.3 ms; SE = 19.3 ms). In this experiment, the category-level congruency effect was observed robustly in the first block of trials, in terms of both accuracy and RT, suggesting that the blue/yellow color-parity associations formed in the first Parity experiment had persisted (panel c of S1 Fig).

The *Congruency* by *Level* interaction was significant, for accuracy, $F_{1,38} = 23.0$, $p < .001$, $\eta_p^2 = 0.38$, and RT, $F_{1,38} = 19.8$, $p < .001$, $\eta_p^2 = 0.34$. The effect of congruency was significant in the category-level version of the experiment, for accuracy, $t_{19} = 4.75$, $p < .001$, $d = 1.06$, and RT, $t_{19} = 5.09$, $p < .001$, $d = 1.14$; but not in the item-level experiment: accuracy, $t_{19} = −1.16$, $p = .13$, $d = −0.26$; RT, $t_{19} = −0.31$, $p = .38$, $d = −0.07$.

The category-level effect was probed for differences between match and mismatch trials (Fig 5c, 5d). There was significantly better congruency performance for match than mismatch trials in terms of accuracy: there was a 4.1% difference

(match: M = 9.17%; SE = 2.15%; mismatch: M = 5.05%; SE = 1.52%), $t_{18}$ = 1.83, p = .042, d = 0.42. There was no significant difference in terms of RT (−9.8 ms difference; match: M = 40.7 ms; SE = 7.67 ms; mismatch: M = 50.6 ms; SE = 13.0 ms), $t_{18}$ = −0.79, p = .22, d = −0.18.

### Post-experiment questionnaire

Participants were asked three questions following the implicit color-number association learning paradigms and explicit association report tasks: 1) What did you notice about the experiment? 2) Did you notice that certain numbers usually appeared in a certain color? If so, please give a description; and 3) What colors did you see during the experiment? Block 1: parity task; Block 2: magnitude task; Block 3: parity task.

The first question was answered by all but four participants (36 participants). About half of these participants spontaneously reported seeing numbers (20) and colors (18). Ten participants (6 in the experimental group) reported noticing a relationship between colors and numbers, while 6 participants (2 in the experimental group) reported an *unreliable* relationship between colors and numbers. Otherwise, some participants commented on: perceived ruptures in perceived stimulus presentation patterns (9); fatigue or difficulty of the experiment (8); making responses (6); the randomization of number or color presentation order (5); and observed changes in number presentation speed/duration (4) and size (3).

In response to the second question, regarding color-number pairings, just over 70% of participants (29 participants) responded yes. In terms of category-level associations, 3 participants correctly reported a relationship between parity and color (blue-even and yellow-odd), and 1 between magnitude and color (green-small and purple-large), in the experimental group (results highlighted in S2 Fig). Strangely, two participants *incorrectly* reported the relationship between parity and color (blue-odd and yellow-even) in the experimental group (one of these participants also incorrectly associated green-even and purple-odd). Three additional participants reported incomplete/inaccurate relationships between parity and color (yellow-odd and blue/purple-even; blue-even; yellow-odd and blue-large); and two participants between magnitude and color (dark/purple-large; blue-large; S2 Fig). One participant in the control, item-level group incorrectly reported a relationship between parity and color; and one participant in the control group incorrectly reported a relationship between magnitude and color (green-small and blue/purple-large). In terms of item-level associations, 15 participants reported specific color-number pairings (6 in the experimental group). On average, these participants reported 2.7 specific color-number pairings (2.6 experimental group; 2.8 control group), with only 48% accuracy (48% experimental group; 47% control group).

In response to the third question, just over half of the participants (21 participants) reported seeing all four colors (yellow, blue, purple, and green). Only 40% of participants (16 participants) correctly reported the combinations of blue-yellow for the parity task and purple-green for the magnitude task. The percent of participants reporting individual colors was as follows: blue (93%); yellow (88%; including one "golden"); green (80%); and purple (73%). Some omissions of the presented colors consisted of substituted colors: for the three participants not reporting blue, all three instead reported seeing green; for the six participants not reporting yellow, two reported seeing orange; for the 8 participants not reporting green, 1 reported seeing blue; and for the 11 participants not reporting purple, 2 reported seeing blue and 1 pink. Three participants reported seeing red (two in the magnitude task), and 1 seeing pink in the parity task. Only one participant commented on the colors in the third part (Parity-Mix) being different (brighter) than in the first part (Parity).

## Discussion

This study had three main aims: 1) to replicate the result that consistency between parity and color categories facilitates implicit learning, relative to corresponding item-level associations of number and color in the context of a parity judgment task [18]; 2) to extend and generalize these results to associations of magnitude and color, in the context of a magnitude judgment task; and 3) to investigate whether there would be an interaction across colors associated with parity and magnitude dimensions. Through using a category-*vs.*-item-level learning paradigm with numbers and colors, we compared

corresponding category-level and item-level experiment versions, each pairing 8 numbers with 2 colors, to test for a potential category-level advantage. Additionally, since numbers can be flexibly categorized simultaneously in multiple dimensions, including parity and magnitude, this design supported probing for potential cross-category associations.

## Category-level associative learning for parity

In regards to our first aim: we found a substantial congruency effect at the category-level for parity, as participants were 7.9% more accurate and 54 ms faster for congruent compared to incongruent trials; Fig 2a, 2b). In the parity-mix experiment, presented to the same participants following the parity (and magnitude) experiments, the parity congruency effect at the category-level was similar: 6.8% accuracy and 45 ms RT (Fig 5a, 5b). In our previous study with the parity paradigm [18], these category-level results were also very similar: an 8.3% accuracy congruency effect accuracy and a 40 ms RT congruency effect. In both the parity and parity-mix experiments, the congruency effect was consistently significant in terms of accuracy and RT (p's ranging from <.001 to 0.006; d's from 0.72 to 1.69); the congruency effect at the category-level was in the predicted direction for both these variables for 85% of participants, and for nearly 70% of participants in the earlier parity study (Fig 3a). Together, these results demonstrate large category-level implicit associative learning effects for parity and color, while participants performed a parity task.

In contrast, at the item-level, there was little evidence of a congruency effect: for parity, −0.1% accuracy and 5 ms RT (Fig 2a, 2b); for the parity-mix experiment, −0.8% accuracy and −1.8 ms RT (Fig 5a, 5b); and for parity in the previous study, −1.5% accuracy and 9 ms RT (i.e., a significant RT effect in the predicted direction: [18]). Across all experiments, there was a significant interaction between *Congruency* and *Level,* for both accuracy and RT. The category-level congruency effect was on average across experiments 8.4% and 42 ms greater than at the item-level. The congruency effect at the item-level was in the predicted direction for both variables in only 20% of participants in this experiment, and 13% of participants in the earlier study (Fig 3a). It may be noted again that despite the lack of strong support for associative learning at the item-level here, the amount of sensory information was matched across the item-level and category-level versions of the experiments, in which 8 numbers (2–9) were consistently paired with 2 colors (blue and yellow). In summary, the results demonstrated a large congruency effect for parity at the category-level but not at the item-level, replicating the study of 2023 [18].

## Category-level associative learning for magnitude

Our second aim was to extend these results to magnitude: we found a significant congruency effect at the category-level for magnitude (accuracy: 4.4%; RT: 37 ms; p's < .001-.008; d's 0.60–1.02; Fig 2c, 2d), with no evidence in favor of a congruency effect at the item-level (accuracy: −1.2%; RT: −9 ms). Once again, the interaction between *Congruency* and *Level* was significant for both accuracy and RT. The congruency effect at the category-level was in the predicted direction for both variables in 65% of participants; and only 5% of participants at the item-level (Fig 3b). The congruency effect was 5.6% and 46 ms greater at the category-level than at the item-level, again, despite an equal amount of sensory information across experiment levels. These results demonstrate that category-level learning with color associations is not specific to parity, but can occur for magnitude as well, suggesting that category-level learning may be a general form of associative learning.

## Category-level associative learning for parity *vs.* magnitude

Although category-level learning occurred advantageously for both parity and magnitude, it is interesting to observe the differences in the results with these two numerical properties, which themselves have fundamental differences. Relative to the category-level results with parity found here and previously, the congruency effect for magnitude was smaller, especially in terms of accuracy. A smaller congruency effect for magnitude than parity was also reported in terms of accuracy in a previous experiment with a color-task (parity effect of 3.1%, relative to only 1.3% for magnitude, and a significant

interaction of level and congruency only for parity; however, the RT effect for parity was only 0.7 ms, relative to 8.7 ms for magnitude: [24]). It is interesting that smaller congruency effects have been found for magnitude than parity, as magnitude is often seen as the most fundamental aspect of numerical processing (e.g., [49,50].

However, the present paradigm was not well-designed to probe this aspect, since participants always performed the parity experiment first (due to the later parity-mix experiment), and with their right hand (dominant in 95% of participants; for comparability with [18]), while performing the magnitude experiment with their left hand, to avoid potential response-level interaction across the parity and magnitude experiment parts. Further, different colors were used for the parity (blue/yellow) and magnitude (green/purple) experiments, possibly affecting the strength of associations formed: blue and yellow were reported to be seen by participants (90% on average) more often than green and purple (76% on average), although this could also be due to the fact that there were two parity (parity and parity-mix) experimental parts. Blue and yellow may also be more strongly opposing colors, relative to green and purple, leading to stronger categorical contrasts ([51,52]). While we did not predict any prior color-category or color-number associations, it is also possible that they did occur (e.g., in the context of data visualization, darker colors are more readily associated with larger quantities: [53]), and future studies could better counter-balance and control associated colors.

Still, one possible theoretical explanation for an advantage for parity in this paradigm is that parity is essentially a binary category that may pair well with two distinctive color categories (potentially relating to representations of symmetry/asymmetry: [54]; although some numbers may be perceived as more even/odd than others, and even may be a more salient category than odd: e.g., [31,55]). In contrast, magnitude can be conceived of as a relative continuum or discrete space, highly dependent on the context of the range of numbers/numerosities presented (e.g., [56–58]). However, a category-level congruency effect for magnitude was found here, perhaps since magnitude can be used to automatically shape binary categories (small/large) in an experimental, i.e., limited context (e.g., small about 20 and large about 50 in [59]; small 1–4 and large 6–9 in [34,60]; small 2–5 and large 6–9 in [24]). Here, small and large numbers were defined from 2–5 and 6–9, in line with reported similarities based on multidimensional scaling ([50,61]; and as in [24]). The discrete vs. continuous nature of magnitude likely also contributes to its categorical nuance: the limited binality of magnitude, and even parity, could be explored in future studies, e.g., using a gradient of purple-to-red, in contrast to a binary grouping, to correspond better with possible magnitude representations.

### Category-level associative learning (with a numerical, rather than color, task)

The effects reported here were much larger than in many previous studies using implicit, statistical associative learning to probe for category-level effects (e.g., categorical word primes and target location: 5 ms [29]; non-word valence: 10–26 ms; 0.5–2.9% [22]; single-exposure categorical words and colors: 2–11 ms; 0.7–1.8% [23]). This is possibly due to the highly probable and repetitive pairing of few stimulus features in this paradigm: with unattended paired numbers' color occurring with very high-probability (~91% congruent; e.g., 75–83% congruent, with several incongruent colors, in [11,16], and with a large number of repetitions (about 500 congruent trial repetitions; Fig 1).

Moreover, at the category-level, there were only 2 levels of parity and 2 levels of color, while at the item-level there were 8 levels of number and 2 levels of color. It may also be that color and number are a good pair for establishing associations, particularly at the category level (e.g., number-color associations in synesthesia, including at the category level with magnitude: [62–65; see [18]). Reasonably prototypical colors were used here, uncontrolled for luminance or saturation, which may also have led to strong memory traces here (e.g., [17,66,67]). While less prototypical colors were used in the parity-mix experiment, and the results were similar to the parity experiment, these relative results are difficult to interpret in light of the order difference (i.e., that the parity-mix experiment was necessarily presented after the parity experiment).

The large effects here are also likely affected by the parity and magnitude tasks used, which directed participants' attention to the relevant conceptual numerical category, while a color task draws participants' attention away from the

numbers and towards the more readily-observable color categories (e.g., [68]). Attention to generalizable stimulus attributes for task demands may enhance category-level learning ([69]; see also [70,71]). Indeed, the effects for both parity and magnitude were also much stronger than in an earlier color-task experiment (3.1% accuracy effect and 0.7 ms non-significant RT effect for parity; 1.3% accuracy effect and 8.7 ms effect for magnitude: [24]).

The numerical tasks here relied on response key presses, which used different keys and hands across the parity and magnitude experiments. These experiments were not specifically designed to probe for response-level learning, such as learning to associate a numerical judgment and/or color with a correlated response key (spatial position) or finger (movement), as has been assessed specifically in previous experiments (e.g., [16,72]). Response keys were not counter-balanced across participants, and color corresponded with the correct congruent responses only in the category-level versions. However, there is no evidence that participants explicitly evaluated color during the implicit associative learning experiments: indeed, only a few participants correctly reported awareness of color/parity or color/magnitude relationships, and the simple numerical tasks were performed rapidly with near-ceiling accuracy for congruent trials, with similar performance in the category- and item-level versions. Importantly, there is evidence that associative learning occurred at a conceptual level in the category-level versions: in the explicit color report task, a different hand and response keys than in the implicit associative learning design were also used, resulting in above-chance accuracy (see the following section). Future studies could attempt to isolate this conceptual associative learning from potential response-level learning effects, for example by changing the response keys across experimental blocks.

### Implicit learning and the explicit color report task

It is interesting to question whether, or to what to extend, the associations learned in the implicit color-number association learning paradigm are really implicit. A post-experiment questionnaire to examine whether associations were formed implicitly was not applied previously with this design with a parity task (i.e., not asked in [18]); however, such a post-experiment questionnaire was added in the subsequent study with a color task ([24]). Here, we applied for the first time the questionnaire to the parity task, as well as to the magnitude task, to further strengthen the validity of this implicit design to investigate category-level associative learning. Nearly all participants were unaware of the category-level manipulations of numerical concepts and color: only 3 participants correctly reported the relationship between parity and color and 1 between magnitude and color. As found here, in previous studies a small portion of participants with awareness of the intended-to-be implicit associations did not appreciably influence the results (S2 Fig; e.g., [13,16,22,24,73,74; but also [17]).

While over two thirds of the participants overall reported noticing color-number relationships when directly asked, their reports were highly variable: at the category-level, 8 participants incorrectly or incompletely reported a parity-color or magnitude-color relationship, including inversely reporting the color and parity relationship. At the item-level, 2 participants falsely reported a relationship between parity and color or magnitude and color, suggesting the numerical task may have cued participants' attention towards these attributes, encouraging them to look for relationships between number and color, even when these relationships were not evident. However, it is also possible that the question itself raised awareness of these attributes that was not consciously attributed during the main experiment: only about a quarter of the participants spontaneously reported a relationship between colors and numbers before being directly asked (e.g., [75]). It may also be that the incongruent trials distracted participants from the high-probability pairings: 15% of participants overall reported an unreliable relationship between colors and numbers.

The more detailed reports of participants were remarkably inaccurate. Nearly 40% of the participants overall (40% of whom were in the experimental group) reported specific pairings between numbers and colors, but with only on average about 3 numbers, and with chance-level accuracy. Further, only about half of the participants correctly reported seeing all four colors in the experiment, with frequent confusion about which colors appeared in which experiment parts: only 40%

accurate overall. Participants' attention was perhaps focused on the numerical attributes, due to the experimental task relating to parity or magnitude, leading to decreased awareness of color, as opposed to passive viewing (see [76]).

In the explicit association report task, participants in the category-level version of the experiment were above chance-level in terms of accuracy at reporting whether the number was associated more with one of two colors, for both parity (67%; Fig 4a) and magnitude (58%; Fig 4c). Participants used a different hand and response keys than in the earlier implicit associative learning experiments, to probe for associative learning beyond the response-level (see [16]). Importantly, this above-chance accuracy does not imply that participants were explicitly aware of the color-number relationships, but may rather provide further evidence for implicit learning [73]. Indeed, the accuracy was not near ceiling: only one participant in the experimental group had perfect association report task accuracy: this participant did not correctly report awareness of the color-parity relationship, but rather incorrectly reported the association between parity and color as yellow/green-even and blue/purple-odd. Overall, more than two thirds of participants had perfect accuracy for only 0–4 of the 8 items (Fig 4e, 4f), again suggesting a lack of explicit awareness of the color-number relationships. Above-chance report of implicit-learning associations has been reported in previous studies [13,16,22,73,74].

The item-level explicit report accuracy was also above chance for parity (63%; Fig 4a), although not magnitude (51%), suggesting that some item-level associative learning may have occurred for parity. This above-chance level accuracy for parity is surprising in that there was a lack of strong evidence for item-level learning for parity, or parity-mix, in terms of the congruency effect in the main experiments: the only congruency effect in the predict direction was a non-significant, 5-ms RT effect in the parity experiment. In the previous parity-task experiment, this congruency effect was of 9 ms, and did reach significance [18]. A limited amount of associative learning at the item-level in the parity experiment potentially relates to the high number of item pairings (i.e., 8, relative to smaller numbers in previous experiments, such as 3–4 pairings in [13,14]. It is likely that more item-level learning would have occurred if more trials or testing sessions were presented (e.g., over multiple testing days: [14–16; see also [77,78]; although there was no hint of an increasing congruency effect across experimental blocks here: S1 Fig). Finally, it is worth noting that the parity explicit report task here was performed after both the parity and parity-mix experiments, and that the parity experiment was always presented first, and so may reflect experimentally-reinforced learning relative to the magnitude report task. On the other hand, the presentation of magnitude as well as parity may have distracted participants from the parity associations: in the previous experiment with only parity [18], an 83% explicit color association accuracy was reported (vs. 67% in the present parity task).

## Interactions between parity-associated and magnitude-associated colors

Finally, our third aim was to investigate whether the colors associated with parity and magnitude (e.g., blue-even and large-purple), might interact across joint numerical categories: specifically, we predicted that the largest congruency effects would be present for parity-magnitude matching colors (e.g., purplish-blue with even-large). Before pursuing this question, it is worth observing that color associations with parity and magnitude persisted through the time of the parity-mix experiment: there was a strong congruency effect in the first block of the category-level parity-mix experiment (S1 Fig), and above-chance associations were reported in the explicit association report task, after all three implicit association learning experiments were completed (Fig 4).

In the parity-mix design, the parity experiment was replicated except that the blue color was shifted toward purple (purplish-blue) and the yellow color was shifted towards green (greenish-yellow). This predicted a parity-magnitude match of color associations for large and even numbers (6 and 8) and small and odd numbers (3 and 5; and mismatching associations for 2,4,7 and 9; Fig 1a, 1b). This was a subtle shift of the colors, as we wanted the blue/yellow associations with parity to remain present. Only one participant commented on the colors in the parity-mix experiment part differing (i.e., being brighter) from those in the parity experiment part. The parity and parity-mix experiments were always separated over time by the magnitude experiment, such that the change in color may not have been consciously observed;

alternatively, participants may have simply not described these within-category color changes in the explicit report task, simply probing what colors were seen. When the stimuli are presented simultaneously the color shift is readily apparent (Fig 1), such that we interpret that the color shift did affect participants' perception.

Indeed, the results were significantly in line with a parity-magnitude match advantage in terms of accuracy, with a matching effect of 4.1%, i.e., a significantly larger congruency effect for matching than mismatching trials (9.2% vs. 5.1%; no RT effect; Fig 5c). This modest matching effect suggests that color associations with conceptual categories may relate to each other. This finding fits into a view in which overlapping numerical conceptual associations can relate to overlapping color associations. Regarding numerical categories, parity and magnitude are interpreted here as fundamental, orthogonal categories (e.g., [50,61]). That is, these categories are non-conflicting, and may be simultaneously referenced without conflict, at least in the case of single-digit numerals. Regarding color categories, we used *adjacent* color category shifts (purple-blue and green-yellow), which can be directly blended together intuitively (e.g., a stimulus can be described as 50% blue and 50% purple without reference to a third color category), since more complex manipulations in constructed color spaces may not be intuitive to observers (e.g., see [79,80]). With the objective of blending adjacent color categories in the parity-mix experiment, we used purple and green for magnitude, instead of replicating the use of red and green for magnitude in a previous study [24].

We propose that associations between conceptual numerical categories and color categories may interact, a similar but more conservative view than that expressed in the speculation that "*all concepts can be viewed as evoking a distribution of associations across all of color space*" (p. 60 [81]; see also [17]). We merely propose that short-term, learned, adjacent color associations may be averaged across stimuli belonging to two orthogonal concepts. One potential, practical extension of this perspective is that if well-known color categories are applied to facilitate learning of novel concepts (e.g., color-coding magnitude symbols for young children; e.g., [35,82]), it might be advantageous to use different color categories for coding magnitude and parity (or other concepts), given potential color-association interactions across dimensions.

## Summary

Associative learning at the category-level, but not item-level, was strongly evidenced in this study, through implicit learning of high-frequency color-number stimulus pairings with a numerical task. This was the case for parity (replicating [18]) and magnitude experiments, with some additional evidence that these parity and magnitude color associations related to each other in a parity-mix experiment. Participants were able to explicitly report the category-level, and item-level parity, color associations above chance-level, with different response keys/hands, suggesting that this associative learning occurred beyond the response-level. Together, these results support that the categorical consistency of numerical concepts facilitates implicit learning of color-number associations. More generally, they suggest that category-level implicit associative learning may be a general principle, operating both within and across multiple associated dimensions.

## Supporting information

**S1 Fig. Performance in terms of accuracy and response time across experimental blocks (5 per experiment part), for Parity (a), Magnitude (b), and Parity-Mix (c).**
(PDF)

**S2 Fig. Experimental data for participants reporting an awareness of color-category associations: participants correctly reporting the color association for both categories are highlighted in pink; participants partially noticing a color association between color and a category are highlighted in orange.** These participants are not outliers, and removing their data has little effect on the pattern of the results.
(PDF)

## Acknowledgments

We gratefully offer thanks to undergraduate student assistants Aurélie Marochi, Laura Klein, Sarah Niesmann, and Nicole Gerasimova, for their fundamental role in data collection. We also thank the journal editor and two anonymous reviewers for their positive but critical comments, which have led to considerable improvement of an earlier version of this manuscript.

## Author contributions

**Conceptualization:** Talia L. Retter, Christine Schiltz.

**Data curation:** Talia L. Retter.

**Formal analysis:** Talia L. Retter.

**Funding acquisition:** Talia L. Retter, Christine Schiltz.

**Investigation:** Talia L. Retter.

**Methodology:** Talia L. Retter, Christine Schiltz.

**Project administration:** Talia L. Retter, Christine Schiltz.

**Resources:** Christine Schiltz.

**Software:** Talia L. Retter.

**Supervision:** Christine Schiltz.

**Validation:** Talia L. Retter.

**Visualization:** Talia L. Retter.

**Writing – original draft:** Talia L. Retter.

**Writing – review & editing:** Talia L. Retter, Christine Schiltz.

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
