## [Decision Letter · Decision Letter 0]

12 Jun 2025

PONE-D-25-18078Categorical consistency of parity and magnitude facilitates implicit learning of color-number associationsPLOS ONE

Dear Dr. Retter,

Thank you for submitting your manuscript to PLOS ONE. After careful consideration, we feel that it has merit but does not fully meet PLOS ONE’s publication criteria as it currently stands. Therefore, we invite you to submit a revised version of the manuscript that addresses the points raised during the review process. I have received two thorough reviews of your manuscript by experts in the field and read the manuscript myself. Both reviewers acknowledge your manuscript addresses an important question and provides valuable empirical evidence. However, several limitations must be addressed before publication. I concur with their evaluation and briefly summarize the main points requiring additional work below. As Reviewer 1 emphasizes, there is a need for a stronger theoretical motivation. Also, Reviewer 2 raises critical methodological concerns about your paradigm's inability to distinguish between response-level and semantic-level learning mechanisms, which should be acknowledged. Both reviewers highlight issues with missing counterbalancing of key experimental factors, which is crucial to discuss further. Additionally, Reviewer 2 identifies redundancies and potential errors in the statistical analyses. The reviewers have provided excellent guidance for these revisions. I encourage you to submit a revised manuscript that addresses these core issues while maintaining the strengths of your empirical work.

We look forward to receiving your revised manuscript.

Kind regards,

Elisa Scerrati

Academic Editor

PLOS ONE

4.  Please expand the acronym “FHSE/UL” (as indicated in your financial disclosure) so that it states the name of your funders in full.

Reviewers' comments:

Reviewer's Responses to Questions

**Comments to the Author**

1. Is the manuscript technically sound, and do the data support the conclusions?

Reviewer #1: Partly

Reviewer #2: Yes

2. Has the statistical analysis been performed appropriately and rigorously? 

Reviewer #1: Yes

Reviewer #2: Yes

3. Have the authors made all data underlying the findings in their manuscript fully available?

Reviewer #1: Yes

Reviewer #2: Yes

4. Is the manuscript presented in an intelligible fashion and written in standard English?

Reviewer #1: Yes

Reviewer #2: Yes

5. Review Comments to the Author

Reviewer #1: “Categorical consistency of parity and magnitude facilitates implicit learning of color-number associations” by Retter and Schiltz:

This manuscript provides valuable insights into the relationship between color and parity and color and magnitude, in terms of whether associative learning can result in implicit associations between these dimensions. The authors conducted a single experiment, with three tasks. The parity task aimed to replicate prior findings by the researchers demonstrating associative learning can emerge between color and parity. The magnitude task was a novel task designed to evaluate whether associative learning could also extend to mapping colors to categories of magnitude. And lastly, the parity-mix task aimed to demonstrate whether learning from tasks 1 and 2 could be merged to influence responses in a final parity task. At the end participants were also asked to complete tasks aiming to explicitly demonstrate learned associations.

The data are appropriately analyzed and the majority of the results are clear and carefully describe key details. There are no concerns about potential competing interests on the part of the authors, data availability, or research ethics. Together, this manuscript reports an interesting experimental design investigating parity/magnitude associations. However, there are several points of concern that I think should be addressed to ensure this work is ready for publication, including clarifying the purpose/context of the work amidst prior work and addressing concerns in methodology of part 3.

Major concerns

1. The theoretical basis/motivation for this work was limited throughout the introduction. There was a clear overview of associative learning and why investigating associative learning and categorical-level (vs. item-level) is important, which was great. However, there was little discussion on why parity and magnitude are beneficial categories to assess in the context of associative learning.

Similarly, it was unclear throughout the introduction what the rationale was for replicating the authors’ previous work on parity, as well as, why it is important to extend these ideas to magnitude. Specifically, I think the work would benefit from additional elaboration and clarification how this this work is different from the previous studies performed by these authors (i.e., adding to the statement on page 2 of the Intro: “Here, we aimed to replicate these results for parity, and extend the paradigm to testing for another category-level association, magnitude, which is a fundamental aspect of number processing.”).

2. The rationale, design, and interpretation of the results of the parity-mix task left me with a number of questions/concerns. The authors mention that all participants completed the tasks in the same order (parity, magnitude, parity-mix), which seems particularly concerning in evaluating the results of this final task. It seems quite possible that order-effects are contributing to the results, such that a primacy effect could be driving why participants were most likely to report blue-yellow colors correctly give those are the color categories first learned in the parity task. This concern could be addressed with additional clarification on the logic for why the order was not balanced across participants and/or, ideally, a set of data in which the order is reversed for tasks 1 & 2.

Further, the colors for the parity-mix task read as an essential component to the task. However, given the limited recognition by the participants that the colors were different, this suggests that the task likely appear the exact same as task 1 for participants. If the colors were more distinct, such that the purplish-blue and yellowish-green were clear border colors between purple and blue and yellow and green (and that participants reported such colors as being on their own color category borders), the results would more directly address the question of whether associations are overlapping across categories.

Further, I wonder whether other kinds of associations could be driving the match vs. mismatched results of this task, such as the dark-is-more bias (e.g., McGranaghan (1989), Schloss, et al. 2018; etc.), in which people infer darker colors represent more. Blues and purples are generally/often darker than yellows and greens and were mapped to larger magnitudes in the present study. Such a bias being activated could provide an alternative explanation to the higher accuracy of matched trials than mismatched. The work would benefit from a discussion considering these points in the context of interpreting the results of the parity-mix task.

3. Throughout the introduction (and a bit beyond too), the use of “category-level” could be interpreted with respect to multiple different dimensions, including parity, magnitude, and color. To clarify, when using “categorical-level”, for the most part (as I understand it), the authors’ use it in reference to the categories of parity and/or magnitude and NOT the color categories used (i.e., the color category of blue). However, in the Introduction’s discussion of the parity-mix experiment, there are references to associations extending to different color categories and in the design of that experiment itself, there is greater consideration color categories (given the colors are shifted towards neighboring color categories) as well as the parity and magnitude categories. The manuscript would benefit from ensuring clarity throughout on which categories are being referenced when discussing the category-level. I encourage the authors to provide additional details and examples throughout the introduction to clarify the multiple dimensions being discussed.

4. In order for others to have the ability to replicate this work, it would be necessary to add information on the following:

a. how much time participants took (on average) to complete each task,

b. how much compensation was given out to participants,

c. how participants were evaluated for atypical color-vision (It seems quite surprising not a single participant of 40 had atypical color vision given prevalence rates)

d. the rationale for the sample size of 40 participants (e.g., was a power analysis performed? If not, how was this number of 20 per condition selected and how is sufficient power ensured?)

5. In the discussion, the authors briefly discuss one practical extension of the work. Aligning with the comments on the introduction and providing additional motivation for why this work is important, it would be beneficial to expand on the implications from these results.

Minor concerns

6. Regarding the methods, it is currently unclear how many total congruent versus incongruent trials were presented for each task. This information seems particularly important to include given the results are dependent on the differences between congruent & incongruent.

A related minor comment, when reporting the results, would help the readers to see the means & variation for each level being used to determine the congruency effect when discussing this effect (versus currently the means/SE are discussed later in the results). This would be especially helpful given that the individual subject data on the results graphs make it a bit challenging to clearly see the means & error bars.

7. While the Supplementary materials are available, a more detailed summary of the main points from those could be incorporated into the main text. Currently, the sentence regarding the “congruency effects were observed to evolve across blocks for the category level experiment….” felt out of place and missing the context needed to understand its importance.

8. There is a bit of inconsistency in the language used throughout the manuscript, that sometimes makes it a bit difficult to interpret the information. For instance, each of the following were used in reference (to at least I think) the same manipulation in the experiment: category-level experiment, category-level version,; experimental group.

9. The discussion had a few typos (the rest of the paper was well-written in that regard): a few examples include; “to” is missing between “compared” and “incongruent” in the first sentence; “leaning” instead of “learning” , “we predicting” instead of “we predicted,” etc.

Reviewer #2: The reviewer comments are in the attached file. The reviewer comments are in the attached file.

The reviewer comments are in the attached file. The reviewer comments are in the attached file.

The reviewer comments are in the attached file.

6. PLOS authors have the option to publish the peer review history of their article (what does this mean? ). If published, this will include your full peer review and any attached files.

**Do you want your identity to be public for this peer review?** For information about this choice, including consent withdrawal, please see our Privacy Policy .

Reviewer #1: No

Reviewer #2: No

---

## [Author Response · Author response to Decision Letter 1]

23 Jul 2025

Reviewers' comments:

Reviewer #1:

“Categorical consistency of parity and magnitude facilitates implicit learning of color-number associations” by Retter and Schiltz:

This manuscript provides valuable insights into the relationship between color and parity and color and magnitude, in terms of whether associative learning can result in implicit associations between these dimensions. The authors conducted a single experiment, with three tasks. The parity task aimed to replicate prior findings by the researchers demonstrating associative learning can emerge between color and parity. The magnitude task was a novel task designed to evaluate whether associative learning could also extend to mapping colors to categories of magnitude. And lastly, the parity-mix task aimed to demonstrate whether learning from tasks 1 and 2 could be merged to influence responses in a final parity task. At the end participants were also asked to complete tasks aiming to explicitly demonstrate learned associations.

The data are appropriately analyzed and the majority of the results are clear and carefully describe key details. There are no concerns about potential competing interests on the part of the authors, data availability, or research ethics. Together, this manuscript reports an interesting experimental design investigating parity/magnitude associations. However, there are several points of concern that I think should be addressed to ensure this work is ready for publication, including clarifying the purpose/context of the work amidst prior work and addressing concerns in methodology of part 3.

Reply: We thank Reviewer 1 for the clear summary and positive evaluation of our manuscript. Please find our responses to the comments below, with changes highlighted in yellow in the accordingly revised manuscript.

Major concerns

1. The theoretical basis/motivation for this work was limited throughout the introduction. There was a clear overview of associative learning and why investigating associative learning and categorical-level (vs. item-level) is important, which was great. However, there was little discussion on why parity and magnitude are beneficial categories to assess in the context of associative learning.

Similarly, it was unclear throughout the introduction what the rationale was for replicating the authors’ previous work on parity, as well as, why it is important to extend these ideas to magnitude. Specifically, I think the work would benefit from additional elaboration and clarification how this this work is different from the previous studies performed by these authors (i.e., adding to the statement on page 2 of the Intro: “Here, we aimed to replicate these results for parity, and extend the paradigm to testing for another category-level association, magnitude, which is a fundamental aspect of number processing.”).

Reply: We fully agree with Reviewer 1 that the motivation for studying associative learning with magnitude and parity should be further elaborated in the introduction. We have substantially modified this section, adding a paragraph as to the interest in studying parity (p. 2, from line 38), and better introducing magnitude in comparison (p. 2, from line 48).

2. The rationale, design, and interpretation of the results of the parity-mix task left me with a number of questions/concerns. The authors mention that all participants completed the tasks in the same order (parity, magnitude, parity-mix), which seems particularly concerning in evaluating the results of this final task. It seems quite possible that order-effects are contributing to the results, such that a primacy effect could be driving why participants were most likely to report blue-yellow colors correctly give those are the color categories first learned in the parity task. This concern could be addressed with additional clarification on the logic for why the order was not balanced across participants and/or, ideally, a set of data in which the order is reversed for tasks 1 & 2.

Reply: Participants did perform the experiments in a fixed order: the rationale for this is expanded in a new paragraph of the revised manuscript (p. 7). Essentially, this was done so that the Parity-Mix experiment could be presented third (after participants had potentially learned both parity- and magnitude-color associations). The Parity (blue/yellow) experiment was then presented first, so that it was not adjacent to the Parity-Mix (purplish-blue/greenish-yellow) experiment, to keep participants’ perception of the shifted Parity-Mix colors from being affected by immediate exposure to the Parity colors. Instead, these two parity experiments were separated by the Magnitude (purple/green) experiment.

In a previous experiment on parity and magnitude, we did counterbalance the order (Retter & Schiltz, 2025, Journal of Cognition), but that was not possible here, in consideration of the additional Parity-Mix experiment. We further discuss the limitations of this fixed order in the discussion, in the context of comparing effects across parity and magnitude categories (p. 15), as well as the explicit report task results (p. 19).

Further, the colors for the parity-mix task read as an essential component to the task. However, given the limited recognition by the participants that the colors were different, this suggests that the task likely appear the exact same as task 1 for participants. If the colors were more distinct, such that the purplish-blue and yellowish-green were clear border colors between purple and blue and yellow and green (and that participants reported such colors as being on their own color category borders), the results would more directly address the question of whether associations are overlapping across categories.

Further, I wonder whether other kinds of associations could be driving the match vs. mismatched results of this task, such as the dark-is-more bias (e.g., McGranaghan (1989), Schloss, et al. 2018; etc.), in which people infer darker colors represent more. Blues and purples are generally/often darker than yellows and greens and were mapped to larger magnitudes in the present study. Such a bias being activated could provide an alternative explanation to the higher accuracy of matched trials than mismatched. The work would benefit from a discussion considering these points in the context of interpreting the results of the parity-mix task.

Reply: We appreciate Reviewer 1’s attention given to the colors used in the experiment. We thought it was important to use shifted colors that still appeared to be within the categories of blue and yellow (with a parity task in this Parity-Mix experiment), since we wanted to preserve the potential blue/yellow associations formed with parity in the first experiment. If we used boarder blue-purple or green-yellow stimuli, this might have imposed new parity-color associations during the course of this third experiment. We now explain the rationale for this parity “color shift” in the revised manuscript (p. 4). We were surprised that only one participant commented on the different colors in the Parity-Mix experiment (and that the color recall report was poor in general): however, we do not think this means that the Parity-Mix colors appeared the same as in the Parity experiment (discussed further on p. 20).

We take the evidence for stronger associative learning for match vs. mismatch numbers as supporting that a (purplish- and greenish-) difference was perceived. In our paradigm, there are two purplish-blue and two greenish-yellow colors in the both the match and mismatch groups, so that we do not think potential prior color associations, such as the dark-is-more bias, could explain these results. However, we take the point that such a bias might affect strength of category-level associations in the magnitude experiment (although, green is actually darker than purple in our stimuli, and we are not actually aware of darkness biases across these categories), and have mentioned this in the revised manuscript (p. 15).

3. Throughout the introduction (and a bit beyond too), the use of “category-level” could be interpreted with respect to multiple different dimensions, including parity, magnitude, and color. To clarify, when using “categorical-level”, for the most part (as I understand it), the authors’ use it in reference to the categories of parity and/or magnitude and NOT the color categories used (i.e., the color category of blue). However, in the Introduction’s discussion of the parity-mix experiment, there are references to associations extending to different color categories and in the design of that experiment itself, there is greater consideration color categories (given the colors are shifted towards neighboring color categories) as well as the parity and magnitude categories. The manuscript would benefit from ensuring clarity throughout on which categories are being referenced when discussing the category-level. I encourage the authors to provide additional details and examples throughout the introduction to clarify the multiple dimensions being discussed.

Reply: We apologize that the term “category-level” was not clear in the manuscript: we use this to describe the experiment version in which associations may occur between numerical (parity or magnitude) and color categories, e.g., between even-blue and odd-yellow. In contrast, the term “item-level” describes the experiment version in which the two colors categories are not consistent with numerical categories (e.g., 2,3,6,7-blue and 4,5,8,9-yellow). We have stated this explicitly in the revised manuscript (p. 1, line 29). We have also included more mentions of both numerical categories and color categories (e.g., p. 2, lines 40-41).

4. In order for others to have the ability to replicate this work, it would be necessary to add information on the following:

a. how much time participants took (on average) to complete each task,

b. how much compensation was given out to participants,

c. how participants were evaluated for atypical color-vision (It seems quite surprising not a single participant of 40 had atypical color vision given prevalence rates)

d. the rationale for the sample size of 40 participants (e.g., was a power analysis performed? If not, how was this number of 20 per condition selected and how is sufficient power ensured?)

Reply: We have added the requested information to the text as follows: a) each implicit associative learning experiment lasted about 15 minutes (p. 5, line 142); the total testing session was 1.5 hours: p. 3; b) participants were paid 15 euros: p. 3; c) participants were only asked about color perception, they were not tested for visual deficits, but normal (or corrected-to-normal) visual acuity and color perception were recruitment inclusion criteria: pp. 3-4; d) the sample size was determined relative to previous implicit association learning studies, particularly the very similar design of Retter, Eraßmy & Schiltz, 2023, PLoS ONE (16 participants per group): p. 3.

5. In the discussion, the authors briefly discuss one practical extension of the work. Aligning with the comments on the introduction and providing additional motivation for why this work is important, it would be beneficial to expand on the implications from these results.

Reply: The discussion section has been expanded, including adding a new beginning section orienting the main experimental aims (p. 13), and a new section on “Category-level associative learning for parity vs. magnitude” (p. 14). We have more carefully addressed some limitations of our design (e.g., not isolating potential response-learning: p. 17), as well as newly emphasized the implications (e.g., last sentence of the summary, p. 21).

Minor concerns

6. Regarding the methods, it is currently unclear how many total congruent versus incongruent trials were presented for each task. This information seems particularly important to include given the results are dependent on the differences between congruent & incongruent.

Reply: We have explicitly added the number of congruent and incongruent trials to the methods (p. 5, line 137).

A related minor comment, when reporting the results, would help the readers to see the means & variation for each level being used to determine the congruency effect when discussing this effect (versus currently the means/SE are discussed later in the results). This would be especially helpful given that the individual subject data on the results graphs make it a bit challenging to clearly see the means & error bars.

Reply: As suggested, we have moved the M and SE reports to the beginning of each results section, when reporting the congruency effect (this change in position is indicated with a yellow highlight only on the beginning of the report, “(congruent: ”, to make it clear that the M and SE data did not change.) pp. 9-11.

7. While the Supplementary materials are available, a more detailed summary of the main points from those could be incorporated into the main text. Currently, the sentence regarding the “congruency effects were observed to evolve across blocks for the category level experiment….” felt out of place and missing the context needed to understand its importance.

Reply: We have added a description of the results in Fig. S1 (p. 10). These results are also now referenced for the Parity-Mix experiment (p. 11; p. 19).

8. There is a bit of inconsistency in the language used throughout the manuscript, that sometimes makes it a bit difficult to interpret the information. For instance, each of the following were used in reference (to at least I think) the same manipulation in the experiment: category-level experiment, category-level version,; experimental group.

Reply: We agree that our language was inconsistent, and have removed all references of “category-level experiment” throughout. It remains only that participants in the experimental group performed the category-level version of each experiment (e.g., p. 1; p. 6).

9. The discussion had a few typos (the rest of the paper was well-written in that regard): a few examples include; “to” is missing between “compared” and “incongruent” in the first sentence; “leaning” instead of “learning” , “we predicting” instead of “we predicted,” etc.

Reply: These typos have been corrected, and we have carefully proofread the discussion. Thank you for this attention to detail.

We again thank Reviewer 1 for the constructive comments, with which we were able to considerably improve our manuscript.

Reviewer #2:

General Assessment

This manuscript presents a series of well-designed experiments investigating how number judgments (parity or magnitude) are facilitated when the numbers are consistently colored according to the target number’s category (odd/even, small/large number) vs. simply consistent at the item level. The authors replicate and extend previous results, finding clear evidence for superiority of associative learning at the category level vs. the item level. The topic is relevant to cognitive psychology, with implications for theories of incidental associative learning and the interpretation and generalizability of it. The manuscript is largely well written, provides all necessary methodological details, adequate analyses and addresses a meaningful research question. However, several aspects of the write-up and interpretation/discussion should be improved before warranting publication.

Major Strengths

1. Convincing Evidence of Category-Level Effects

The consistent and statistically robust findings for category-level congruency effects in accuracy and RTs strongly support the idea that this association is superior to mere item-level associations.

2. Replication and Extension

The study replicates prior findings and meaningfully extends them to magnitude

associations, which strengthens the reliability and generalizability of the reported effects.

3. Extensive Post-Experiment Questionnaires

The systematic probing of subjects for learned associations with different response keys/hand allows for a thorough picture on the explicitness of the presented associations.

Reply: We thank Reviewer 2 for the positive evaluation, including highlighting these strengths of our manuscript.

Major Weaknesse

---

## [Decision Letter · Decision Letter 1]

24 Aug 2025

Categorical consistency of parity and magnitude facilitates implicit learning of color-number associations

PONE-D-25-18078R1

Dear Dr. Retter,

We’re pleased to inform you that your manuscript has been judged scientifically suitable for publication and will be formally accepted for publication once it meets all outstanding technical requirements.

Kind regards,

Elisa Scerrati

Academic Editor

PLOS ONE

Additional Editor Comments (optional):

Reviewers' comments:

Reviewer's Responses to Questions

**Comments to the Author**

1. If the authors have adequately addressed your comments raised in a previous round of review and you feel that this manuscript is now acceptable for publication, you may indicate that here to bypass the “Comments to the Author” section, enter your conflict of interest statement in the “Confidential to Editor” section, and submit your "Accept" recommendation.

Reviewer #1: All comments have been addressed

Reviewer #2: All comments have been addressed

2. Is the manuscript technically sound, and do the data support the conclusions?

Reviewer #1: Yes

Reviewer #2: Yes

3. Has the statistical analysis been performed appropriately and rigorously? 

Reviewer #1: Yes

Reviewer #2: Yes

4. Have the authors made all data underlying the findings in their manuscript fully available?

Reviewer #1: Yes

Reviewer #2: Yes

5. Is the manuscript presented in an intelligible fashion and written in standard English?

Reviewer #1: Yes

Reviewer #2: Yes

6. Review Comments to the Author

Reviewer #1: This revised manuscript has substantially improved in clarity and rigor from the original submission. Thank you to the authors for their work to carefully and meaningfully address the questions and concerns that were originally posed. The introduction now provides a stronger motivation for why the authors are studying parity & magnitude. Throughout, the article is more clear with respect to terminology (e.g., “category level”), their rationale for design decisions (i.e., the decision to not counterbalance the task orders), and potential alternative explanations/limitations of the work. The authors also added essential details about their study, which would be needed for future replication by others (e.g., timing, compensation, trial numbers, etc.).

My one comment to note (which may be intentional by the authors), although the authors provide a URL to their data on Zenodo, the URL link (at least at this time) appears to link to a broken page (i.e., “DOI not found” message). Apologies if this is intentional (i.e., if the data will be publicly posted upon publication). I did not see the mention of such qualifications, so wanted to make mention that the URL currently does not permit access to the data, in case this is due to a typo/error for the URL link.

Otherwise, in my reading of the revised manuscript and the authors’ response to the reviews, I believe the present work is suitable for publication in PLOS One.

Reviewer #2: Thank you for your detailed responses. I am satisfied with the result and find the manuscript much improved.

7. PLOS authors have the option to publish the peer review history of their article (what does this mean? ). If published, this will include your full peer review and any attached files.

**Do you want your identity to be public for this peer review?** For information about this choice, including consent withdrawal, please see our Privacy Policy .

Reviewer #1: No

Reviewer #2: No

---

## [Editor Report · Acceptance letter]

PONE-D-25-18078R1

PLOS ONE

Dear Dr. Retter,

I'm pleased to inform you that your manuscript has been deemed suitable for publication in PLOS ONE. Congratulations! Your manuscript is now being handed over to our production team.

Kind regards,

on behalf of

Dr. Elisa Scerrati

Academic Editor

PLOS ONE